# Purification of Myosin from Bovine Tracheal Smooth Muscle, Filament Formation and Endogenous Association of Its Regulatory Complex

**DOI:** 10.3390/cells12030514

**Published:** 2023-02-03

**Authors:** Lu Wang, Isabel J. Sobieszek, Chun Y. Seow, Apolinary Sobieszek

**Affiliations:** 1Centre for Heart Lung Innovation, St. Paul’s Hospital, University of British Columbia, Rm 166-1081 Burrard Street, Vancouver, BC V6Z 1Y6, Canada; 2Department of Medicine, University of British Columbia, Vancouver, BC V5Z 1M9, Canada; 3Division of Pediatric Nephrology and Gastroenterology, Department of Pediatrics and Adolescent Medicine, Comprehensive Center for Pediatrics, Medical University of Vienna, 1090 Vienna, Austria; 4Department of Pathology and Laboratory Medicine, University of British Columbia, Vancouver, BC V6T 2B5, Canada; 5Formerly associated with the Austrian Academy of Sciences, 1010 Vienna, Austria

**Keywords:** airway smooth muscle, asthma, myosin filaments, myosin regulatory proteins

## Abstract

Dynamic regulation of myosin filaments is a crucial factor in the ability of airway smooth muscle (ASM) to adapt to a wide length range. Increased stability or robustness of myosin filaments may play a role in the pathophysiology of asthmatic airways. Biochemical techniques for the purification of myosin and associated regulatory proteins could help elucidate potential alterations in myosin filament properties of asthmatic ASM. An effective myosin purification approach was originally developed for chicken gizzard smooth muscle myosin. More recently, we successfully adapted the procedure to bovine tracheal smooth muscle. This method yields purified myosin with or without the endogenous regulatory complex of myosin light chain kinase and myosin light chain phosphatase. The tight association of the regulatory complex with the assembled myosin filaments can be valuable in functional experiments. The purification protocol discussed here allows for enzymatic comparisons of myosin regulatory proteins. Furthermore, we detail the methodology for quantification and removal of the co-purified regulatory enzymes as a tool for exploring potentially altered phenotypes of the contractile apparatus in diseases such as asthma.

## 1. Introduction

In hollow organs that undergo large volume changes, the smooth muscle lining the walls readily modifies its contractile function in order to suit the changing geometry. A striking property of smooth muscle myosin is its ability to be assembled and disassembled rapidly in an intact, functioning smooth muscle cell. This evanescence of myosin filament allows the muscle to swiftly rearrange its contractile units and adapt to large changes in muscle cell length [1], thereby maintaining optimal physiological functions [2]. Evanescence of myosin filaments has been proposed as one of the key mechanisms responsible for the proper functions of airway smooth muscle (ASM) and its alteration responsible for the pathological features in some airway diseases [3]. It has been suggested that overly stable myosin filaments (i.e., lack of evanescence) may contribute to the pathophysiology of asthma [4,5]. An increased Rho kinase protein content reported in asthmatic airways [6] may lead to increased stability of myosin filaments, resulting in the lack of response of asthmatic airways to the bronchodilatory and bronchoprotective effect of deep inspiration [7,8,9,10,11].

Smooth muscle myosin molecules in solution are known to self-assemble into filaments, and the stability of the filaments can be tested in vitro [12]. One critical piece of missing information is the structural stability of myosin filaments from asthmatic ASM compared with that of non-asthmatic ASM. To better understand the dynamics of myosin, it is useful to establish an effective methodology for isolating functional smooth muscle myosin from the airways.

While the properties of smooth muscle myosin in vitro are largely modified by the ionic conditions [13] and pH [14], they are well regulated in vivo by the cellular milieu, resulting in the characteristic malleability of the filaments in comparison to that of skeletal muscle. It was originally shown that the solubilized smooth muscle myosin molecule may adopt one of two possible configurations: the folded one (a loop) being more soluble, or the extended (straight) one, less soluble, aggregating to dimers and tetramers [15,16]. Such tetramers are associated with coiled-coil rods, forming what is considered in vivo the myosin filament-building units [17].

In skeletal muscle, contraction is primarily regulated by the troponin regulatory complex, which includes a Ca^2+^ binding subunit. By contrast, there is no troponin in smooth muscle [18]. Instead, its analogous protein calmodulin (CaM), which has four Ca^2+^ binding sites, can respond to changing intracellular Ca^2+^ concentration by binding to Ca^2+^. The resulting Ca^2+^- CaM complex activates the myosin light chain kinase (MLCK). MLCK, in turn, phosphorylates the 20 kDa regulatory myosin light chain (ReLC) to initiate a contraction [19,20,21,22,23]. Relaxation requires dephosphorylation of the ReLC by myosin light chain phosphatase (MLCP). Although other modulatory proteins such as telokin (TL) and structural proteins such as filamin or caldesmon are associated with myosin filaments, Sobieszek and colleagues [24,25] concluded that the regulatory complex is composed of CaM, MLCK, and MLCP and that this regulatory complex is bound to (and is co-purified with) the myosin filaments. Another group of researchers has also pointed out that MLCK and CaM co-purified with smooth muscle myosin due to their tight association with myosin rather than actin [26]. Depending on buffer composition, various degrees of MLCK can be copurified and/or removed from myosin [27]. The employment of a reliable procedure for the purification of smooth muscle myosin and the study of the regulatory complex is of primary importance to the understanding of normal physiological functions and possible dysfunctions of ASM.

This manuscript describes a purification protocol that was originally developed for gizzard smooth muscle by Sobieszek and Bremel [18], an approach in which most of the associated proteins are removed before the extraction of the targeted myosin and the regulatory complex bound to it. This protocol was later applied to pig stomach muscle [19] and successfully optimized for bovine tracheal smooth muscle [12]. In the present report, we comprehensively present the gradient sodium dodecyl sulfate polyacrylamide gel electrophoresis (SDS PAGE) [28] and the urea glycerol polyacrylamide gel electrophoresis (UG PAGE) [29], which can be used to document protein content of the many extracts obtained at the different stages of the procedure and to study ReLC phosphorylation of purified myosin, respectively. Besides providing a review of the methodology for myosin extraction from intact ASM tissue, this report aims to bring insight into the regulatory enzymes that are usually co-purified with myosin. The quantification and removal of co-purified regulatory proteins provide additional tools for exploring potentially new phenotypes of the contractile apparatus in diseases such as asthma.

## 2. Evolvement of the Sobieszek–Bremel Purification Approach for Smooth Muscle Myosin (from Gizzard to Airway Smooth Muscle)

Smooth muscle contains myosin and actin, both tethered with their regulatory proteins and enzymes. In order to purify it, myosin must be separated from all other components. To find the best way to separate out or purify myosin while preserving its biological function and characteristics is the basis of the evolvement of the Sobieszek–Bremel purification approach over the years.

Sparrow and colleagues [30] were the first to develop a method to extract actomyosin that maintained some sensitivity to Ca^2+^. However, the extracted actomyosin exhibited relatively low ATPase activity. A remarkable advancement in the approach and procedures of myosin purification was achieved by Sobieszek and Bremel [18]. These authors obtained purified myosin from a crude actomyosin by precipitation at a high MgCl_2_ concentration followed by precipitation in ammonium sulphate (am.sulf.) (Table 1). The essence of this approach is to create the condition for myosin and actin to be both in filamentous form and dissociated. This is accomplished by extensive fragmentation (mincing) of the smooth muscle tissue followed by thorough homogenization at low ionic strength and removal of the non-contractile proteins by repetitive washing and centrifugation. Crude actomyosin is then extracted from the resulting myofibril-like preparation (MYF) in the presence of ATP and calcium-chelating agents (EDTA and EGTA). The ATPase activity of the crude actomyosin is relatively high and Ca^2+^ sensitive, representing a suitable source for myosin purification, namely a simple precipitation at high MgCl_2_ concentration [22].

In a follow-up study, it was determined that divalent cations alone were effective in causing actomyosin precipitation overnight; the obtained actomyosin was 4–5 times more sensitive to Ca^2+^ than that obtained with am.sulf. precipitation, although the high Mg^2+^ concentration appeared to have determined a reduced actin-activated ATPase activity [31]. In a study using pig stomachs, the actomyosin obtained using 2 mM Ca^2+^ precipitation showed high ATPase activity, but very low sensitivity to Ca^2+^ [19]. The authors tested various modifications to the original Sobieszek–Bremel method in pig stomachs but did not detect substantial improvements. Tested modifications included total precipitation of the crude actomyosin with am.sulf. (55–60% saturation), precipitation with Ca^2+^ or Mg^2+^ together with am.sulf. (60% saturation), and extraction at intermediate or high ionic strength (0.12 and 0.5 M KCl, respectively).

In a further modification of the method described in a study using turkey gizzard, Bis Wash (BW) buffer was used in place of the “washing buffer” of the original recipe to obtain MYF, followed by low ionic strength actomyosin extraction solution (LAMES) to obtain crude actomyosin. Actin was then removed from crude actomyosin to yield pure myosin [23]. This protocol was successfully applied to bovine tracheal smooth muscle [12]. The evolvement of the purification methods is detailed in Table 1. The general scheme of the purification process is depicted in Figure 1. Modification and improvements from the original method have been described in the subsequent reviews by Small and Sobieszek [20] and Sobieszek [23]. The scheme in Figure 1 of the present report is an extension of previously published schemes to include the latest observations and improvements, specifically for bovine tracheal smooth muscle. During purifications of myosin from gizzard muscle and pig stomach, we had no difficulty in collecting the assembled filaments after dialysis against BW solution, provided that the dialysis was sufficiently long and included two or three changes in the BW. Dialysis in the cold room was terminated when precipitation became visible at the top part of the dialyzing bag; despite some “loss” of myosin in the supernatant, the amounts of purified myosin were always large. However, this was not the case for the MYF from bovine tracheal smooth muscle. In the case of the airways, the amount of starting material is limited, thus requiring further improvement of the protocol to minimize the loss of myosin in the process of purification. It is one of the purposes of this manuscript to describe an extended purification scheme that is suitable for ASM.

In Figure 2, we include an example of SDS PAGE image to show the protein composition and relative content in myofibrils from turkey gizzard and bovine tracheal smooth muscle. For convenience, the lane of bovine tracheal smooth muscle has also been included as a reference lane (R) in Figure 3, Figure 4, Figure 5 and Figure 6, where the purification yields and their approximate composition are shown in the subsequent lanes of SDS PAGE. The sample in R is different from those of the other lanes, which are also from bovine tracheal smooth muscle. Different samples obtained from large (above 150 g), small (2–3 g), and very small amounts (about 1 g) of muscle tissue were used for Figure 3, Figure 4 and Figure 5, respectively. Each figure shows the progression of the same sample as it undergoes the steps of the purification method depicted in Figure 1.

There are important technical steps in the Sobieszek protein purification method for smooth muscle myosin that must be taken into consideration. When starting from the whole muscle, it is important to dissect smooth muscle free of connective tissue and fat, so that the preparation contains as little as possible contamination of non-muscle tissue. The presence of large amounts of connective tissue in the muscle preparation makes myosin extraction difficult. The presence of filamentous actin can also significantly modify the physical and biochemical properties of myosin preparation [23]. Therefore, the purity of the starting tissue is of great importance to ensure the quality and quantity of extracted myosin.

After dissection, the muscle can either be used fresh or frozen before the homogenization step. If used fresh, the muscle should be minced extensively, since, as noted in the original publication for this procedure [18], the mincing step ultimately determines the size of the fibril bundles. Freezing the dissected muscle in liquid nitrogen (LN_2_) [12] allows for pulverization of the muscle tissue, which has the advantage of achieving better contact of the tissue with the homogenization solution as well as optimal preservation of the biochemical properties of myosin. Whether to use minced or frozen tissue also depends on the amount of preparation. For minimal portions (grams) of muscle, the use of frozen and then pulverized tissue yields better results; for large amounts of muscle, mincing fresh tissue may be a more appropriate way to facilitate homogenization. A mincer equipped with a special 1.25 mm-hole plate is a great option for large amounts of tissue when considering the losses of material involved. After mincing, we recommend using the Sorvall type omni-mixer homogenizer, because of its very effective cutting edges of the blades, and a tight glass–glass Dounce-type homogenizer. Two or more cycles of homogenization, centrifugation, and resuspension (Table 1) produce the MYF-like preparation from which crude actomyosin is extracted.

## 3. Characterization of the Purified Myosin from Bovine Tracheal Smooth Muscle

### 3.1. Composition of Purified Bovine Tracheal Smooth Muscle Actomyosin

Although large volumes of extraction buffer and several repeated extractions in succession are required to completely extract crude actomyosin from smooth muscle MYF, a single extraction, with the extraction buffer volume at five times that of the muscle myofibril, yields a considerable portion of the actomyosin in high concentration (up to 25 mg/mL). The predominant or readily identified species in the extracted smooth muscle crude actomyosin are, besides myosin with its regulatory (Re) and essential (Es) light chains (LCs), actin and the actin-binding proteins tropomyosin (TM), α-actinin, and filamin. These and the other identifiable proteins are shown in Figure 2. Other proteins in the crude actomyosin extract need further purification steps to be identified as bands on the gel. These proteins include the regulatory MLCK and its activator CaM, as well as MLCP. A high-molecular-weight protein normally not entering our gradient gels can be also recognized during myosin purification of the bovine tracheal smooth muscle (see Figure 4, lane L, and Figure 5, lane H). This most likely corresponds to the smitin discovered by Kim and Keller [32]. The role of smitin in smooth muscle is not currently understood, although it can be hypothesized, based on its large titin-like size, that it interacts with myosin and may be involved in the myosin filament assembly. We suggest that the observed smearing or reduced penetration of the myosin heavy chains into the top stacking gel containing urea could be due to the presence of smitin in the purified myosin.

### 3.2. Regulatory Light Chain (ReLC) Phosphorylation of Purified Myosin

Regulatory light chain phosphorylation of the purified myosin can be estimated from UG PAGE first introduced by Sobieszek and Jertschin [29], or determined exactly by ^32^P incorporation from radioactive ATP. For exact determination, the phosphorylation reaction is initiated in the presence of 0.1 mM CaCl_2_ by the addition of an ATP solution in which γ^32^P-ATP is only present at a negligible concentration in comparison to “cold”, unlabeled ATP. The mixed suspension is quenched in a solution of 8.5 M urea and 40 mM 4-(hydroxymercuril) benzoic acid with vortex. The quenched mixture is spotted on 2 × 4 cm pieces of Whatman 3MM chromatographic paper and processed for radioactivity counting. The pieces are then dropped into a 10% TCA solution and extensively washed with several changes of water to remove the non-incorporated radioactive ATP. The advantage of this determination is in the use of water as a convenient “scintillant” (Cherenkov radiation!) for the counts per minute instead of the flammable organic scintillant commonly used for radioactivity determination [33]. Another advantage of using radioactive ATP is the relatively short decay time of the radioactive ^32^P, since it only requires a few months of storage for the radioactive waste to be safely disposed of as a non-radioactive one.

In the case of the parallel ^32^P incorporation and UG PAGE determination of the ReLC phosphorylation, the reaction is terminated by the addition of 8.5 M urea as previously described [12]. The ReLC has a high phosphorylation rate, therefore it can only be determined by this method at 0°C (on ice) when the phosphorylation rate is reduced by 10-fold in comparison to that at 37 °C. At 0 °C, the phosphorylation reaches a maximum within 30–60 s from the starting of the phosphorylation reaction, which then slowly declines as a result of dephosphorylation by the presence of MLCP. Dephosphorylation is enhanced by the depletion of ATP as well as by the residual ATPase activity of the myosin alone [12]. The essential light chains (EsLCs) can be recognized on UG PAGE gels to migrate as a doublet because of their different charges (Figure 6 and Figure 7).

It is also possible to evaluate myosin phosphorylation from standard (non-radioactive) ATPase activity assay (AA) with phosphate determination. In this case, after the termination of the reaction with 10% TCA solution, the normally discarded and denatured small protein pellet can be resuspended in water and pelleted again to be dissolved in the urea/2-mercaptoethanol solution and processed as for the UG PAGE [34]. As shown in Figure 7, the ReLC is phosphorylated by the endogenous CaM/MLCK complex. The second band in Figure 7 can be interpreted as ReLC being phosphorylated at the first and/or second sites, while the third band can be interpreted as phosphorylation occurring at the third and/or fourth sites based on the notion of ordered or sequential phosphorylation described by Persechini and Hartshorne [35].

### 3.3. Formation of Filamentous Myosin by Dialysis

Purified myosin molecule retains its secondary and tertiary structure. Visualized under an atomic force microscope, non-phosphorylated myosin molecules in a solution of high ionic strength (500 mM) clearly show the head and tail regions [12]. Myosin filaments form when the ionic strength is lowered, especially when ReLC phosphorylation is not inhibited. The filaments can be formed by dialyzing a solution of myosin monomers (1.3 mg/mL) at a high ionic strength relaxing solution (composition in mM: 5 EGTA, 1 Mg^2+^, 5 MgATP, and 1- PIPES, sufficient KCl to produce 500 mM ionic strength, pH adjusted to 7.0 at 25 °C) against a buffer containing 2 mM MgCl_2_ over 17 h at 4 °C during which the ionic strength is decreased linearly and gradually from 500 to 88 mM.

Alternatively, purified myosin at a concentration of 2 μg/mL can readily form filaments in activating solution at low ionic strength (80 mM) (composition in mM: same as the above relaxing solution with the addition of 100 nM microcystin, 2 mM CaCl_2_, and 5 mM ATP to induce ReLC phosphorylation). The formation of filaments confirms that the C-terminal end of the coiled–coil domain, i.e., the critical 28 residues responsible for myosin filament formation [36] is completely intact. The co-purified or native-like CaM/MLCK complex facilitates the phosphorylation of the ReLC, which promotes myosin filament assembly [37] and increases filament stability, such that the filaments do not disassemble even in the presence of ATP [14]. When visualized under an atomic microscope or an electronic microscope, myosin filaments formed in relaxing solution appear to have larger diameters than those formed in activating solution, though filaments formed in activating solution appear in greater numbers [12]. Furthermore, the filaments formed in activating solution are much more resilient than filaments assembled in relaxing solution when perturbed with ultrasonic agitations, thus confirming that when ReLC is phosphorylated the filament stability is increased [12].

## 4. Myosin Filament Assembly and Copurification of the Regulatory Complex of Smooth Muscle

There is compelling evidence that CaM, MLCK, and MLCP form a functional complex in smooth muscle [24,25,33]; this is also the case for the myosin purified from tracheal smooth muscle. Their co-purification or native-like localization on myosin filaments is perhaps determined by their physiological function pertaining to the formation and stability/maintenance of the filaments. It has been suggested that these tightly associated regulatory enzymes may act in vivo producing phosphorylation (or dephosphorylation) of the myosin ReLC by actively moving along the myosin filaments [23].

It is known that MLCK and MLCP share the common substrate ReLC. They regulate the phosphorylation of ReLC, which is critical in filament formation and/or stabilization. After adding MgATP and Ca^2+^, MLCK immediately phosphorylates ReLC in the presence of CaM, also co-purified in the myosin preparations. Freshly purified myosin becomes fully phosphorylated on Ser19 within 10–15 s even when kept on ice (0 °C) [12]. It appears that in ASM, myosin was purified in a phosphorylated state, and was co-purified with the endogenous functional complex composed of the regulatory complex. This is shown in Figure 7 by their phosphorylation at the first and/or the second site, in the presence of 0.1 mM CaCl_2_, after the addition of MgATP. Phosphorylation at the third site is also seen in Figure 7. Maximal phosphorylation of myosin at all four sites was not observed. Due to MgATP depletion, ReLC dephosphorylates within 3–5 min. This demonstrates the presence of co-purified MLCP in the myosin preparations and that the filamentous myosin can undergo a reversible phosphorylation process that may correspond to the contraction–relaxation cycle of intact smooth muscle.

Besides main regulators MLCK and MLCP, other important proteins also modulate smooth muscle ReLC phosphorylation levels. Examples are CaM and TL. CaM acts as an activator of the MLCK, and TL acts as an inhibitor. TL slows down the phosphorylation produced by MLCK and accelerates the inhibitory effect of MLCP, both located on the myosin filaments [38]. TL also reduces the affinity of MLCK for myosin filaments, thereby reducing the myosin phosphorylation rate but not the level of phosphorylation [38]. In the absence of Ca^2+^ or at its nanomolar range of concentration, CaM/MLCK becomes inactivated [39,40]. Under physiological conditions, MLCK exists in several oligomer forms [41], with the activated form having a lower molecular weight and the inactivated form having a higher molecular weight. CaM is not directly involved in the oligomerization process, but, in its presence, MLCK undergoes autophosphorylation [42], and the smaller molecular weight oligomer of MLCK shows greater affinity to CaM [39].

## 5. Quantification of the Endogenous Kinase Complex in Myosin

The molar ratio of the endogenous CaM/MLCK complex to myosin is 1:30 or less [43]. MLCP co-purifies with myosin but at an approximately 20–50-fold lower stoichiometric ratio than MLCK [23]. Detailed molar ratios of the endogenous MLCK, CaM, and MLCP in the native-like filamentous myosin preparations can be found in Table 1 of Sobieszek et al. [38].

The use of SDS PAGE allows separation and identification of the main contractile proteins involved such as TM, CaM, MLCK, and the catalytic and targeting subunits of MLCP [24,25]. However, these endogenous complexes cannot be detected on heavily loaded electrophoresis gels of myosin. Nevertheless, they may be readily detected by their very high activities even at 0 °C, and have been clearly seen using gel filtration chromatography [44].

The addition of high concentrations of either MLCK or CaM to the freshly formed filamentous myosin does not result in any significant increase in the endogenous rate of ReLC phosphorylation, which indicates CaM and MLCK are present in the filaments in an approximately stoichiometric ratio. In the presence of micromolar concentrations of Ca^2+^, when MLCK and/or CaM are added separately or together, the phosphorylation rate may increase or stay unchanged. This phenomenon can be used to estimate the content of the CaM-MLCK complex. The rate of ^32^P incorporation of myosin can be detected in the presence of increasing amounts of both enriched CaM and/or MLCK using higher sample loadings. The absolute amount of endogenous MLCK is then estimated from the linear range of the rate increase in the presence of added known amounts of MLCK saturated with CaM [28]. With excessive sample loading, CaM can also be detected using UG PAGE, since its migration also depends on its saturation with Ca^2+^. The molar amounts of CaM and MLCK tightly bound to myosin filaments are approximately the same [33]. The concentration of MLCK and CaM varies with purification conditions and perhaps species but is sufficient to phosphorylate the myosin component within 15–30 s (assay rate) following the addition of Ca^2+^ (50–100 μM) and MgATP (0.5–1.5 mM) [43].

## 6. Removal of the Native-like Regulatory Complex from the Purified Smooth Muscle Myosin

### 6.1. Tropomyosin Removal from MYF and Filament Formation

As a result of the high ionic strength during am.sulf. fractionation, TM is released from the thin actin-containing filaments and does not precipitate with the latter but co-precipitates with myosin at the higher am.sulf. saturations (Figure 1). TM is also soluble at low salt concentrations. Therefore, its removal is relatively simple, except for a portion that is trapped in the relatively large am.sulf. pellet of myosin. The am.sulf. salt in the pellets solubilizes myosin and is removed during the subsequent dialysis and filament assembly. Thus, non-soluble filaments can form and be pelleted by centrifugation; only traces of TM might be present in the purified filamentous myosin. The amount of trapped TM depends on the volume ratio between the supernatant and the pellet, the latter being small at this stage of the process [22,28]. TM traces can be clearly identified on SDS or UG PAGE and can be removed by a simple resuspension of the filaments in the BW buffer followed by an additional pelleting.

### 6.2. Purification of CaM, MLCK, and MLCP

Calmodulin is purified from the supernatant of the first (or together with the second) wash after the removal of TM and other abundant proteins at 60% am.sulf. saturation. CaM is then precipitated with 85% saturated am.sulf. and after pelleting, it is dissolved in a low ionic strength buffer containing 0.5 mM EGTA. After dialysis and removal of insoluble proteins, 0.5 mM CaCl_2_ and 15% am.sulf. are added, and the clarified supernatant is loaded onto a Phenyl Sepharose affinity column (Pharmacia, Uppsala Sweden) equilibrated with the same buffer at 4 °C. After extensive washing of the column, CaM is eluted with the same buffer in which Ca^2+^ is replaced by 2 mM EGTA. The presence of CaM is monitored by UG PAGE due to the characteristic shift of its band after the removal of Ca^2+^. CaM in the pooled fractions should be concentrated by isoelectric point precipitation at pH 4.1 or placed into dialysis bags for a concentration/precipitation step in which solid am.sulf. is placed outside these bags to reduce the internal volume and precipitate CaM. The CaM-containing pellet is then collected, solubilized, and subjected to gel filtration on an AcA 54 column (LKB, France). A homogeneous preparation of CaM is obtained after a subsequent ionic exchange/purification step. This is followed by another concentration/precipitation step with solid am.sulf. The dissolved pellet is dialyzed against AA buffer (in mM: 60 KCl, 2 MgCl_2_, 0.5 DTT, and 10 imidazole, pH 7.4 at 4 °C) and stored at -30 °C [33,39].

The extraction and purification of MLCK step is accomplished by including an additional washing or extraction with kinase extraction solution (KES, Figure 8) of high MgCl_2_ concentration during the MYF preparation. This is accomplished before the crude actomyosin step [20,37,45] and the MYF preparation can be continued by the two additional washes to remove the MgCl_2_ trapped in the pellet. The main reason for using a high concentration of MgCl_2_ is to increase the ionic strength needed for MLCK extraction and at the same time to significantly reduce the amount of the co-extracted TM. High concentrations of TM in the MLCK extracts result in high viscosity, thus precluding further purification of the kinase [24,25]. Performing the intermediate extraction of MLCK with KES results in distinct crude actomyosin and subsequent types of myosin. The content of co-purified or endogenous CaM/MLCK complex is 3–4-fold lower with the MLCK extraction step than without it [38].

It is our experience that MLCP and MLCK always co-purify together to varying extents. To obtain a fraction that contains solely the high molecular weight MLCP, myofibrils are first subjected to 40–55% am.sulf. fractionation which yields the MLCK-MLCP complex. The complex is subjected to a gel filtration step on an AcA34 column (5.0 × 95 cm) in AA buffer. The fractions containing MLCK and MLCP activities elute together at approximately 350 kDa [24,25]. This kinase–phosphatase complex is run through a pair of chromatography columns (strong cation and anion exchanger) connected in tandem. After extensive washing, the two columns are separately eluted. Most of the MLCK is bound by the first column. Practically all the MLCP is bound to and eluted from the second column [24,25].

### 6.3. Removal of the Regulatory Complex from Myosin

In order to obtain myosin free of CaM/MLCK, the CaM/MLCK complex is most conveniently removed directly from the crude myosin (CMY) am.sulf. pellet, which is soluble in BW as a result of the salt trapped in the large pellets. Am.sulf. also increases the hydrophilicity and enhances the binding of CaM and MLCK to the gels used in their removal. After adding 0.2 mM CaCl_2_ and 0.6 M NaCl_2_ to the solubilized CMY, the solution is passed through CaM-affinity and Phenyl Sepharose affinity columns (Pharmacia, Uppsala Sweden) connected in tandem and equilibrated with the same buffer [33]. The resulting CMY is free of CaM/MLCK and can be further purified as the other two myosin types described above (with or without the KES step). The three types of myosin preparation were described by Sobieszek [37] and characterized by Sobieszek [27].

The supernatant fraction obtained by the precipitation of crude actomyosin with MgCl_2_ is determined to be rich in MLCK tightly bound to myosin [46]. After raising the pH to 7.6 to 8.0 to precipitate some of the residual tropomyosin, this supernatant is cleared by centrifugation and then dialyzed against the AA stock solution (in mM: 60 KCl, 1 MgATP, 1 cysteine, 40 imidazole, pH 7.0 at 25 °C) to remove inorganic phosphate. The insoluble components are finally removed by centrifugation to yield MLCK [46]. MLCK can also be obtained by direct extraction of myofibrils in the absence of ATP. This extraction is facilitated at higher ionic strength (μ = 0.12) and at alkaline pH (7.4 to 7.8) with the inclusion of MgCl_2_ to minimize extraction of actin and tropomyosin. The composition of the extraction medium is the same as the AA stock solution except with pH adjusted to 7.4 to 7.8 at 4 °C and with added 15 to 30 mM MgCl_2_. The extract is dialyzed against AA stock solution to precipitate insoluble material and the kinase is purified by gel filtration in the same buffer on Sepharose 6B-CL (Pharmacia, Uppsala, Sweden) or Sephadex G150 [33,39].

## 7. Application in Asthma Research

It has been demonstrated that if a sufficiently large stretch is applied to the ASM, it can result in transient depolymerization of myosin filaments and temporary reduction in force-generating ability of ASM [47,48] This unique character, referred to as myosin filament evanescence, may at least in part underlie the mechanisms for one of the hallmark features of asthma. Lack of deep inspiration-induced bronchodilation [49] and/or bronchoprotection [50] has been consistently demonstrated in asthmatic patients, while both phenomena appear to be the first line of defense against bronchospasm in healthy subjects [51]. Measurements of the mechanical properties of asthmatic ASM reveal the same response to simulated deep inspiration as that displayed by asthmatic patients [52], i.e., deep inspiration not only fails to provide relief, but may also further exaggerate the existing bronchospasm. These observations point to the direct relevance of myosin filament stability to bronchoconstriction in asthma.

We have shown that myosin filament stability in ASM tissue is regulated by Rho-kinase [4] and that Rho-kinase protein content is significantly elevated in ASM from asthmatics compared with non-asthmatics [6]. Rho-kinase activity has also been shown to increase in cultured human tracheal smooth muscle cells from asthmatic compared with non-asthmatic subjects [53]. Rho-kinase is known to promote ReLC phosphorylation by inhibiting MLCP. As demonstrated by Ip et al. [12], phosphorylated myosin filaments, formed by purified myosin following the methodology described above, are much more resistant to mechanical agitations. These observations have led to a postulation that the intrinsic labile properties of myosin molecule as well as factors that regulate myosin filament formation and stability hold the key to our understanding of asthma pathophysiology and can potentially provide new treatment strategies for asthma. For example, we have proposed the inhibition of Rho-kinase coupled with therapeutic pressure oscillations to synergistically depolymerize myosin filaments as a new treatment strategy for asthma [5].

To understand the physiological vs. pathological properties of myosin in vivo, it may be helpful and necessary to obtain the molecule in as much native state as possible using the techniques provided by this review. Advanced techniques such as proteomic or mass spectrometry can be certainly used to characterize the purified myosin and the isolated regulatory enzyme complex. In addition, these techniques have the potential to detect and discriminate structural and sequence alterations in the purified myosin and regulatory complex that are caused by diseases such as asthma. Combined with these downstream techniques, studies of purified myosin and its regulatory enzymatic complex in vitro could provide clues of the molecular regulations for myosin filament evanescence underlying healthy and diseased states.

In addition, asthma is characterized by the exaggerated contraction of ASM. Phosphorylation of ReLC, catalyzed by MLCK, plays a critical role in ASM contractility. MLCK has been shown to be elevated in ragweed-sensitized canine ASM [54,55] and hyperresponsive guinea pigs [56]. It has been shown that smooth muscle MLCK is increased in sensitized human bronchi [57] It has also been shown that both mRNA and protein expression of MLCK are increased in human asthma [58]. There has been no report on the quantification of MLCK that is bound to myosin. Since it is the regulatory enzymes residing within the smooth muscle contractile filament lattice that are responsible for the direct regulation of ReLC phosphorylation and ASM contraction, it is of interest to investigate whether the amount of MLCK and MLCP bound to myosin filament are also altered in asthma. As we have shown above, the presence of regulatory enzymes co-purified with myosin can be quantified [23,38]. There is potential in using their relative quantity (e.g., the ratio of MLCK/MLCP to myosin) as a biomarker for ASM contractility in asthmatic vs. non-asthmatics patients for following disease progression and understanding disease mechanisms. The filament stability test developed by Ip et al. [12] can also be used to test the stability of myosin filaments purified from ASM from asthmatic and non-asthmatic subjects.

## 8. Conclusions

In this review, the Sobieszek–Bremel [18] approach for purification of ASM myosin is presented in detail with emphasis on the presence of co-purified proteins TM, CaM, MLCK, and MLCP, their physiological relevance, and the method for their removal to obtain pure myosin. Although the goal for purification is to obtain the protein to understand its structure, function, and factors that regulate its activities, it is important to recognize the tight association between myosin and the co-purified regulatory proteins such that they are ever-present unless specific steps are taken for their removal. The quantification of the co-purified regulatory proteins could potentially hold a key in improving our understanding of the mechanisms of asthma and can serve as a biomarker for identifying new phenotypes of the disease and disease progression.

## Figures and Tables

**Figure 1 cells-12-00514-f001:**
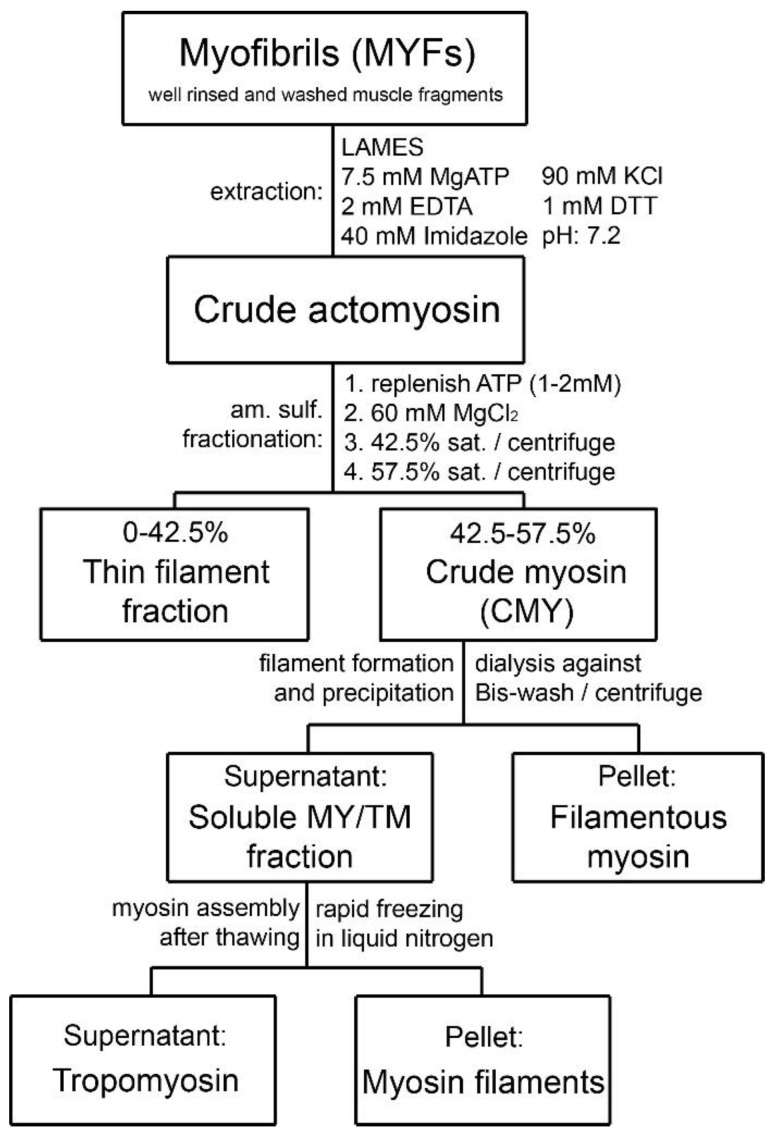
Schematic diagram of the purification steps developed for bovine tracheal smooth muscle myosin starting from isolated myofibrils. The approach was originally developed for gizzard muscle [18].

**Figure 2 cells-12-00514-f002:**
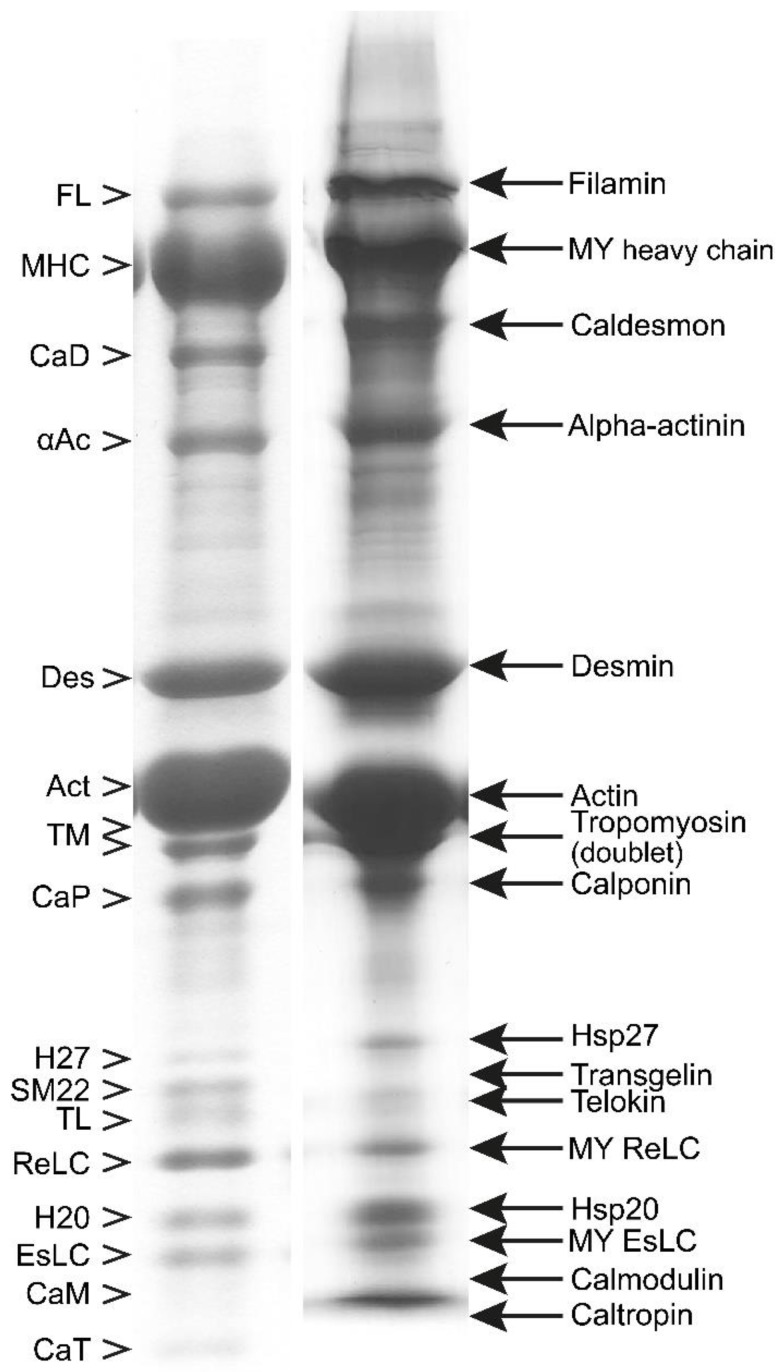
Identification of the major proteins of smooth muscle myofibrils by a gradient SDS PAGE. Left lane: turkey gizzard. Right lane: bovine trachea. The arrows on the right-hand side point to the individually labeled proteins. The empty arrowheads on the left-hand side point to the same proteins labeled with abbreviated names. In the right lane, calmodulin and caltropin overlap with the dye-front band.

**Figure 3 cells-12-00514-f003:**
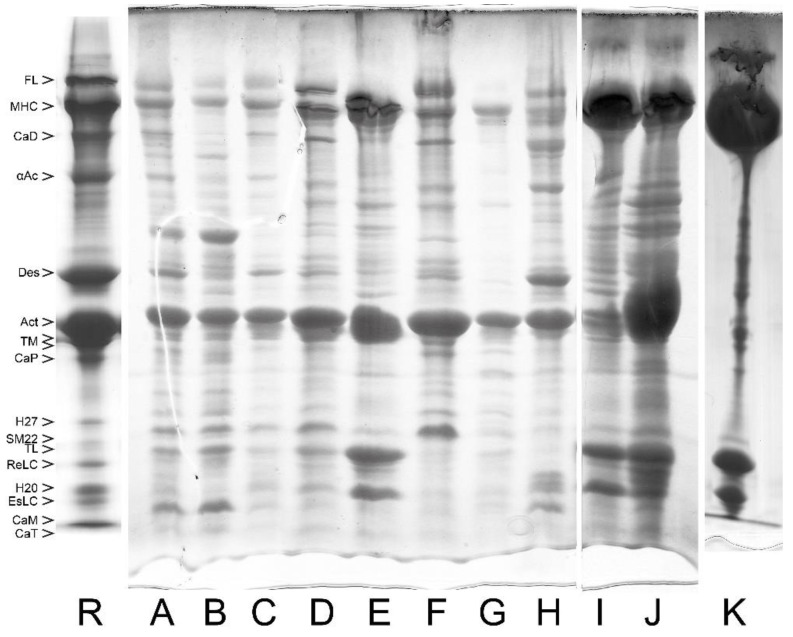
Purification steps of bovine tracheal smooth muscle myosin analyzed by SDS PAGE system for a large amount of starting muscle tissue (above 150 g). These and the following gels were obtained using a gradient SDS PAGE system [28]. Individual bands are identified and labeled with abbreviated names. (R) Reference lane taken from the right lane of Figure 2. (**A**) Whole muscle. (**B**) First wash in BW solution of the whole muscle. (**C**) Myofibrils (MYF). (**D**) LAMES extract (crude actomyosin). (**E**) CMY (40–55% am.sulf. pellet of crude actomyosin dissolved and clarified). (**F**) Thin filaments fraction (40% am.sulf. pellet of crude actomyosin). (**G**) CMY pellet dissolved in AA solution and clarified by centrifugation. (**H**) The residue obtained after LAMES extraction. (**I**) Self-assembled myosin filaments collected by centrifugation and resuspended in AA solution. (**J**) Heavily loaded soluble myosin fraction, or the supernatant of lane I. (**K**) Purified myosin obtained after intermediate freezing in LN_2_, pelleting, resuspension/washing in BW, and further pelleting and resuspension in AA. Note that in this and all following figures, the wordings “clarified” or “pellet after clarification” mean procedural removal of insoluble proteins that were not used in the present study.

**Figure 4 cells-12-00514-f004:**
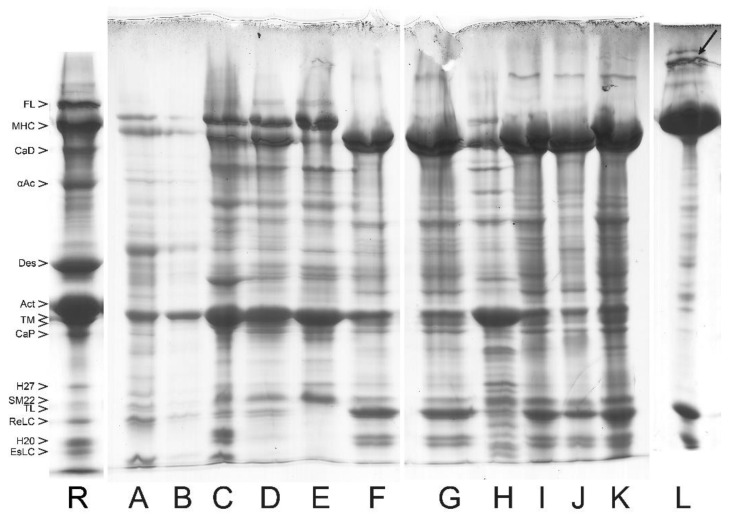
Purification steps of bovine tracheal smooth muscle myosin analyzed by SDS PAGE system for a small amount of starting muscle tissue (2–3 g). Individual bands are identified and labeled with abbreviated names. (R) Reference lane taken from the right lane of Figure 2. (**A**) First wash showing significant loss of soluble myosin. (**B**) Second wash in 2-fold diluted BW without noticeable loss of myosin at the 2-fold higher loading (8 µL) on the gel. (**C**) Myofibrils (MYF). (**D**) Crude actomyosin extracted with LAMES. (**E**) Thin filaments fraction of crude actomyosin (40% am. sulf. pellet). (**F**) CMY pellet of crude actomyosin (40–57.5% am.sulf. fraction); note the clearly separated TM doublet, which is also visible in G. (**G**) CMY pellet after solubilization in BW and clarification. Note the presence of some actin and tropomyosin, mostly “trapped” in the large-volume pellet. (**H**) Pellet obtained after clarification of the CMY shown in lane G. (**I**) CMY after myosin filament assembly step (o/n dialyzing against BW solution). (**J**) Myosin filaments pelleted and resuspended in AA solution. (**K**) Supernatant after collection of the filaments shown in J. (**L**) Purified myosin obtained after rapid freezing and storing of the supernatant (lane K) by LN_2_ and its subsequent fast thawing. The black arrow at the top of the lane indicates smitin, which co-purifies with myosin to some extent. Note the presence of TL in most of the lanes, just above the band of ReLC.

**Figure 5 cells-12-00514-f005:**
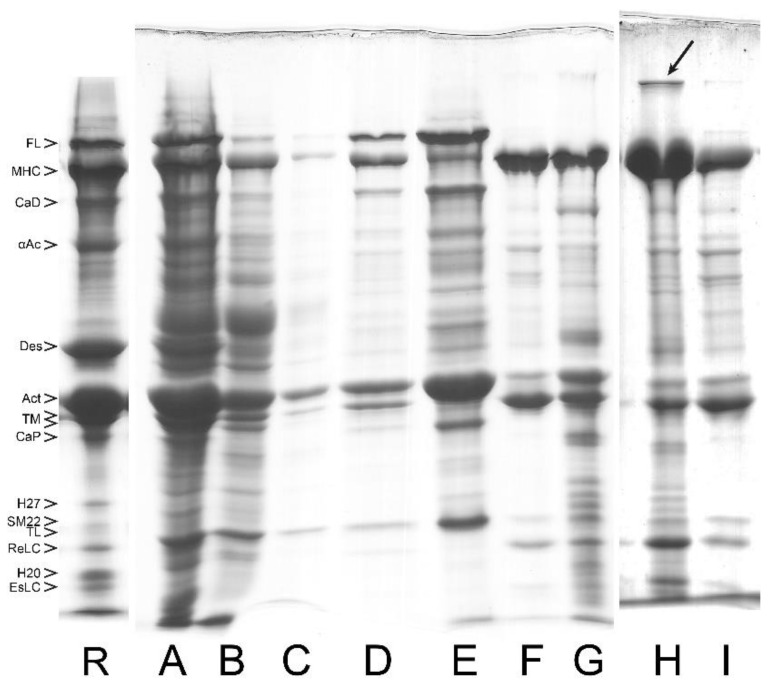
Purification steps of bovine tracheal smooth muscle myosin analyzed by SDS PAGE system for a very small amount of starting muscle tissue (about 1 g). Individual bands are identified and labeled with abbreviated names. (R) Reference lane taken from the right lane of Figure 2. (**A**) Whole bovine tracheal smooth muscle homogenized in BW with Polytron in a 2 mL Eppendorf tube. (**B**) Supernatant of the first wash. (**C**) Second wash. (**D**) LAMES extract (crude actomyosin). (**E**) Thin filament fraction of crude actomyosin (40% am.sulf. pellet). (**F**) CMY (40–57% am.sulf. pellet of crude actomyosin). (**G**) Myosin pellet (CMY dialyzed o/n against BW). (**H**) Subsequent pellet after freezing in LN_2_. The black arrow at the top of the lane indicates smitin, which co-purifies with myosin to some extent. (**I**) Supernatant of H. This figure shows that the loss of myosin (even under BW conditions) is considerable, which leads to low yield of purified myosin (see lanes **H**,**I**). It is advisable to use larger starting material whenever possible.

**Figure 6 cells-12-00514-f006:**
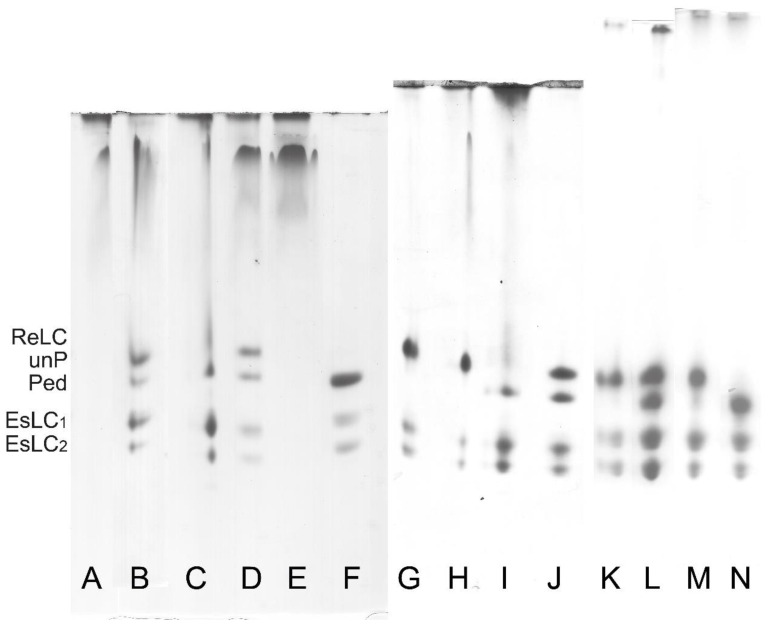
Analysis of phosphorylation levels of ReLC by UG PAGE of three different myosin preparations. As indicated by the labels on the left, the top bands correspond to unphosphorylated (unP) ReLC. The bands below the unphosphorylated bands correspond to phosphorylated (Ped) ReLC. The bottom two bands correspond to the two differently charged isoforms of EsLC, which are not phosphorylated. The first preparation (lanes **A**–**E**) was bovine tracheal smooth muscle myosin purified using the standard protocol developed for gizzard smooth muscle. The second preparation (lanes **G**–**J**) was bovine tracheal smooth muscle myosin purified using the protocol optimized for bovine tracheal smooth muscle. The third preparation (lanes **K**–**N**) was pig stomach muscle myosin purified using the protocol optimized for bovine tracheal smooth muscle. (**A**) Crude actomyosin. ReLC and EsLCs did not enter the separation part of the gel containing glycerol because they were not dissociated from the myosin heavy chains [29]. (**B**) Freshly prepared CMY supernatants. (**C**) CMY pelleted filaments. In lane B and C, ReLC and EsLC entered the gel but with significant smearing. (**D**) CMY supernatant left on ice overnight. ReLC and EsLC migrated into UG PAGE and the characteristic TM smearing band is visible at the top of the lane. (**E**) Supernatant of D after pelleting of the myosin filament. The only band visible in lane E is TM because the soluble TM remained in the supernatant after the removal of the myosin filaments. (**F**) Phosphorylated purified myosin from turkey gizzard included for comparison. (**G**) Freshly prepared CMY supernatant. (**H**) Pelleted filaments of the dialyzed CMY fraction. (**I**) CMY supernatant left on ice overnight. (**J**) Supernatant of I after pelleting of the myosin filament. Analogous purified myosin preparations were obtained from other types of smooth muscle, i.e., the antrum phasic (lanes **K**,**L**) and fundus tonic (lanes **M**,**N**) muscles of pig stomach.

**Figure 7 cells-12-00514-f007:**
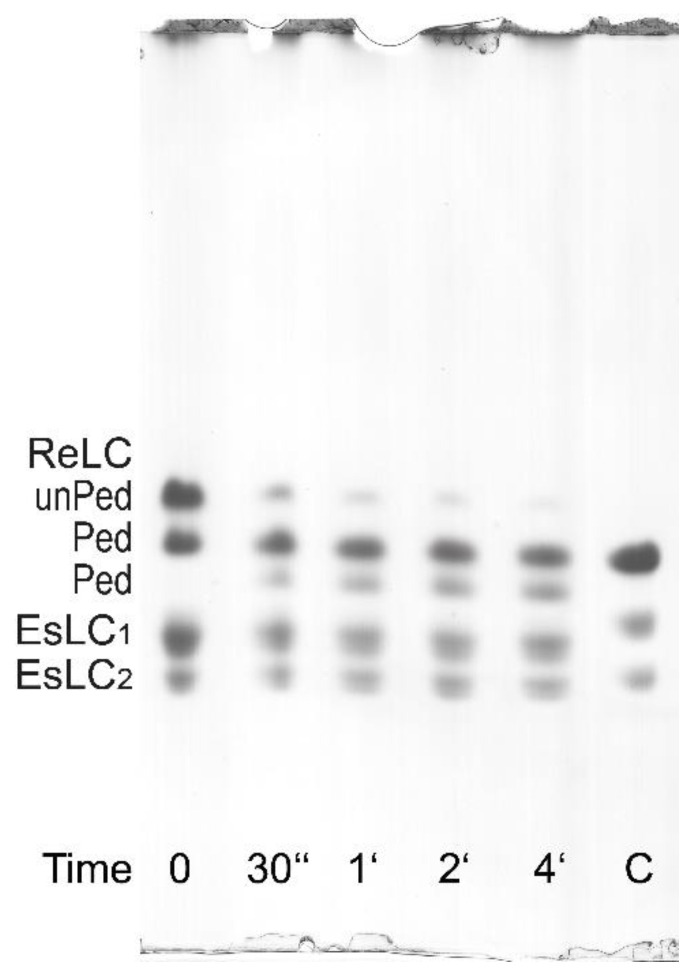
Phosphorylation assay of the purified bovine tracheal smooth muscle myosin ReLC on UG PAGE. As indicated by the labels on the left, the top bands correspond to unphosphorylated ReLC (unPed); the second and the third bands correspond to phosphorylated ReLC (Ped). The lower doublets correspond to pairs of the EsLCs and are present in all lanes. The phosphorylation assay was performed at 0 °C (on ice) by adding MgATP in the presence of 0.1 mM CaCl_2_. The decreasing intensity of the top bands and the increasing intensity of the third bands show the presence of the endogenous CaM/MLCK complex. For comparison, a phosphorylated turkey gizzard myosin (lane **C**) is included.

**Figure 8 cells-12-00514-f008:**
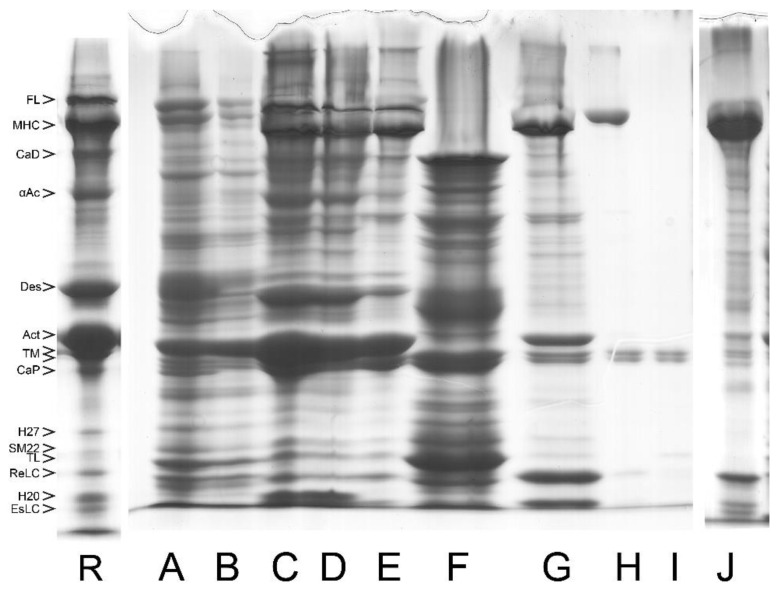
Purification of myosin from bovine tracheal smooth muscle with intermediate extraction of MLCK regulatory complex with kinase extraction solution (KES). Such extraction procedure has been routinely used for purification of the regulatory complex and TL [24,25]. (R) Reference lane. (**A**) First wash of the whole tissue. (**B**) Second wash. (**C**) MYF after the KES extraction. (**D**) Third wash to remove the KES solution. (**E**) LAMES extract or the “alternative crude actomyosin”. (**F**) KES extract as used for MLCK purification. (**G**) CMY (the alternative 40–57.5% am. sulf. pellet of crude actomyosin) in AA. TM doublet is clearly visible. (**H**) Myosin loss or the TCA precipitate of the 57.5% am.sulf. supernatant of G. (**I**) TM as the only content (TCA precipitate) of the 57% am.sulf. supernatant of F. (**J**) Purified myosin obtained after the CMY was dialyzed o/n against BW and the filaments pelleted and resuspended in a small volume of AA.

**Table 1 cells-12-00514-t001:** The evolvement of Sobieszek myosin purification approach.

References	Source	To Obtain Myofibril (MYF)		To Obtain Actomyosin	To Remove Tropomyosin	To Obtain Pure Smooth Muscle Myosin
		Mincing	Homogenization	More homogenization	Am.Sulf. Precipitation	Actin Removal and Further Purification
Sobieszek and Bremel 1975 [18] Sobieszek 1977 [22]	Chicken or turkey gizzard.	Twice with 1.2 mm holed meat grinder; this step determines the size of the fibril bundles.	“Washing buffer”: KCl (60 mM), cysteine (1 mM), MgCl_2_ (1 mM), imidazole (20 mM), and streptomycin (100 mg/L), pH 6.9 at 4 °C, Triton X-100 0.3–0.5%. Homogenization for 20 s with a Sorvall Omnimixer. Two passes with a Teflon-glass homogenizer. Centrifugation (12,000× *g* 10–15 min).	ATP (10 mM), EDTA (1 mM), KCl (60 mM), cysteine (1 mM), MgCl_2_ (1 mM), imidazole (40 mM), and streptomycin (100 mg/L), pH 7.1 at 4 °C. Centrifugation (12,000× *g* 30 min). Filtration through glass wool.	Am.sulf. (25% saturation). Centrifugation 12,000× *g* 45 min. Redissolvation of pellet in a solution containing KCl (60 mM), cysteine (1 mM), and imidazole (20 mM) pH 7.0.	Centrifugation at 40,000× *g* 16 h to yield crude myosin fraction. Precipitation with 35–40 mM of either CaCl_2_ or MgCl_2_. Redissolvation and dialyzation against 0.3–0.6 M KCl to remove Ca^2+^. Dilution to 60 mM KCl in the presence of ATP (1 mM) and EDTA (1 mM) pH 7.0. Dialyzation overnight against a medium containing 0.6M KCl, EGTA (1 mM), imidazole (20 mM), pH 7.0, and centrifugation at 105,000× *g* 2.5 h.
Sobieszek and Small 1976 [31]	Chicken or turkey gizzard.	Same as above.	Same as above.	Mg^2+^ precipitation (pH 6.8).	Same as above.	Same as above.
Small and Sobieszek 1977 [19]	Pig stomach.	Same as Sobieszek and Small 1976 [31].	As above for gizzard, except pH 6.8 and addition of 250 µM of the PMSF inhibitor and 2–3 passes in the Glenco 300 mL size Teflon-glass tight homogenizer.	Same as Sobieszek and Small 1976 [31], except pH 6.9, CaCl_2_ (2–20 mM).	Same as Sobieszek and Small 1976 [31].	Same as above for gizzard, but for pig stomach and bovine tracheal smooth muscle purifications, it was important to add also Pefabloc S C protease inhibitor (0.1 µM final) to the dissolved CMY just before the o/n dialysis.
Sobieszek 1994a [23], Ip et al. 2007 [12]	Bovine trachea.	Dissected muscle pulverized in LN_2_.	“Bis Wash buffer (BW)” (in mM: 40 KCl, 2 MgCl_2_, 10 imidazole, 0.5 DTT, 0.5% Triton X-100, and 10 Tris base) pH 6.6 at 4 °C. Centrifugation 39,100× *g* 30 min.	Pellet homogenization in “LAMES” (in mM: 90 KCl, 2 EDTA, 2 EGTA, 7.5 Na_2_ATP, 1 DTT, and 40 imidazole) pH 7.2 at 4 °C.	Am.sulf (40–60% saturation).	ATP replenishment and an excess of MgCl_2_ (50 mM). Am. sulf. (40% saturation, then 60% saturation). Pellet dissolvation in BW and dialyzation against 50% BW overnight then full BW for an additional 3 h. Centrifugation at 29,325× *g* 30 min at 4 °C and the pellet resuspension in BW.

## Data Availability

Data is contained within the article.

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
