# Peer review of "Purification of Myosin from Bovine Tracheal Smooth Muscle, Filament Formation and Endogenous Association of Its Regulatory Complex"

_cells, 2023, doi:10.3390/cells12030514_

Round 1

Reviewer 1 Report

It is great to see a group of scientists improving methodology that appears to be basic science, but actually would be important for the detection in diseases. A few comments for the manuscript:

1)      Line 33: maybe don’t start the Introduction with ‘in contrast’. Needs a line or two before this sentence as a general introduction.

2)      Line 38: move the reference to the end of the sentence.

3)      Line 39: abbreviate and replace ‘smooth muscle’ as ‘ASM’ here, instead of Line 49.

4)      Line 58: replace ‘and’ with ‘or’.

5)      Line 87: does the ‘sample’ refers to ASM from different diseases?

6)      Line 111: is CAM a common abbreviation for crude actomycin? This may confuse the readers as CaM is also mentioned in the manuscript.

7)      Line 131-134: perhaps consider rearranging or rewriting the flow of these sentences as the readers might forgotten that ‘this scheme’ in Line 134 was mentioned in Line 131.

8)      Table 1: is it necessary to write ‘same as above for gizzard’? Can you just put ‘same as above’? or can you just say same as ‘Sobieszek 1977’?

9)      Table 1: what are the underlines mean?

10)   Figure 2: are both lanes bovine tracheal sample or is one lane supposed to be smooth muscle from another organ?

11)   Figure 2: what does ‘18 kDA protein?’ means?

12)   Figure 3: are the different lanes belong to the same sample (A) and B-L is showing that sample progression as it goes through purification? If yes, perhaps be clearer about it in Line 154-156 and Figure 3 legend. And in Line 154-156, be clear how Figure 3-5 differs as readers might missed the difference between these figures. Perhaps you could add the differences in the steps between Figure 3-5.

13)    Figure 5: does it mean you cannot use sample <2 g?

14)   Line 212: how do you monitor?

15)   Line 222: maybe indicate ‘mincing fresh tissue’ as readers might forgot mincing is for fresh tissue.

16)   Line 242: what do you mean by ‘non-recognizable’?

17)   Line 246: can you point out which band in Figure 4L and 5H?

18)   Please never start a sentence with an abbreviation.

19)   Line 271: maybe add ‘phosphorylation’ in ‘because of its high phosphorylation rate’.

20)   Figure 6 is not cited in-text? And also is this showing the same sample as it progresses in the protocol?

21)   Line 308: there is no lane P or O?

22)   Line 309: what’s the significance of phasic and tonic as they appeared suddenly?

23)   Figure 8 also not cited in-text.

24)   Line 326-327: which method is preferred?

25)   Line 516: compared ‘with’ not compared ‘to’.

26)   Line 541: the close bracket is missing.

27)   Subheading for section 2: maybe can somehow indicate the evolvement towards the detection in ASM?

28)   Section 5 and 6: not sure if I am missing the point/difference, but should section 6 comes first (removal and purification) then only section 5 (quantification)?

29)   Is it possible to have a figure for Section 6?

30)   Line 523: can you speculate or suggest what type of treatments?

Author Response

Comments and Suggestions for Authors

It is great to see a group of scientists improving methodology that appears to be basic science, but actually would be important for the detection in diseases. A few comments for the manuscript:

We would like to thank the reviewer for the insightful comments and suggestions that have greatly helped improve our manuscript. We have made all the corrections. Please note the new line numbers refer to the line numbers when track changes are in place as required by the journal for revision submission.

1)      Line 33: maybe don’t start the Introduction with ‘in contrast’. Needs a line or two before this sentence as a general introduction.

Response 1: We have removed the phrase “in contrast to skeletal muscle” to focus only on smooth muscle. We have added an introductory sentence as suggested.

2)      Line 38: move the reference to the end of the sentence.

Response 2: We have moved the reference to the end of the sentence (line 43).

3)      Line 39: abbreviate and replace ‘smooth muscle’ as ‘ASM’ here, instead of Line 49.

Response 3: We have abbreviated the ASM here (line 42) as suggested and replaced the “airway smooth muscle” on line 54 (previously line 49) with “ASM”.

4)      Line 58: replace ‘and’ with ‘or’.

Response 4: We have replaced “and” with “or” (now line 62).

5)      Line 87: does the ‘sample’ refers to ASM from different diseases?

Response 5: No, the word “samples” here refers to the yielded product after each step of the purification procedure. To avoid confusion, we have replaced the word “samples” with “extracts” (now line 96).

6)      Line 111: is CAM a common abbreviation for crude actomycin? This may confuse the readers as CaM is also mentioned in the manuscript.

Response 6: We have removed the abbreviation “CAM” for crude actomyosin and kept the abbreviation “CaM” as it is commonly known for calmodulin. We have made this correction throughout the manuscript text, Table 1, Figure 1, and figure legends.

7)      Line 131-134: perhaps consider rearranging or rewriting the flow of these sentences as the readers might forgotten that ‘this scheme’ in Line 134 was mentioned in Line 131.

Response 7: The phrase “This scheme” refers to the schemes in previously published review papers. The scheme has been now extended in the present manuscript. We have modified the sentence to say “The scheme in Fig.1 of the present report is an extension of previously published schemes to include….” (lines 145-146). 

8)      Table 1: is it necessary to write ‘same as above for gizzard’? Can you just put ‘same as above’? or can you just say same as ‘Sobieszek 1977’?

Response 8: We have now used “same as above” as suggested.

9)      Table 1: what are the underlines mean?

Response 9: We have now removed all underlines.

10)   Figure 2: are both lanes bovine tracheal sample or is one lane supposed to be smooth muscle from another organ?

Response 10: The two lanes were duplicates of the same sample. We erroneously thought they were from a bovine tracheal smooth muscle whole tissue sample. Upon double-checking, we realized that the sample was in fact a turkey gizzard myofibril sample. Prompted by the reviewer’s question, we thought it would be a good idea to have a comparison between different organs. Therefore, we have kept one of the two lanes and added a new lane from a bovine tracheal smooth muscle myofibril sample for comparison.

11)   Figure 2: what does ‘18 kDA protein?’ means?

Response 11: After looking into this, we realize that this is the band of “HSP20”. We have now replaced “18kDA protein” with “HSP20” in the new Figure 2.

12)   Figure 3: are the different lanes belong to the same sample (A) and B-L is showing that sample progression as it goes through purification? If yes, perhaps be clearer about it in Line 154-156 and Figure 3 legend. And in Line 154-156, be clear how Figure 3-5 differs as readers might missed the difference between these figures. Perhaps you could add the differences in the steps between Figure 3-5.

Response 12: No, A and B-L were not from the same sample. Now we have replaced lane A with a new reference lane (the new bovine tracheal smooth muscle myofibrils lane in Fig 2. See response 10 above). Lanes B-L are now lanes A-K, which show sample progression as it goes through purification. They are from the same sample but different from lane R (reference lane). How Figure 3-5 differ is the different amount of starting material. We have clarified this in the figure legend and currently lines 164-174.

13)   Figure 5: does it mean you cannot use sample <2 g?

Response 13: Although Figure 5 shows that myosin purification is possible with samples < 2 g, due to the unavoidable protein loss during the purification process, it is preferable to use larger samples. We have modified the legend of Figure 5 (lines 242-243) to reflect this point.   

14)   Line 212: how do you monitor?

Response 14: We have replaced the “carefully monitored” to “is of great importance to” (line 251).

15)   Line 222: maybe indicate ‘mincing fresh tissue’ as readers might forgot mincing is for fresh tissue.

Response 15: We made the correction as suggested (line 262).

16)   Line 242: what do you mean by ‘non-recognizable’?

Response 16: We meant to say these proteins do not appear as identifiable bands on the gel when loading crude actomyosin extract. We now modified the sentence to say “Other proteins in the crude actomyosin extract need further purification steps to be identified as bands on the gel. These proteins include…”(line 284).

17)   Line 246: can you point out which band in Figure 4L and 5H?

Response 17: The band is the one at the very top. We have indicated this band with a black arrow in the new figure 4 lane L and figure 5 lane H. The protein name (Smitin) is added to the legends of both figure 4 (line 226) and figure 5 (line 239).

18)   Please never start a sentence with an abbreviation.

Response 18: We now spelled out ReLC as “regulatory light chain” to start the paragraph (line 297). We also made similar corrections throughout the rest of the manuscript.

19)   Line 271: maybe add ‘phosphorylation’ in ‘because of its high phosphorylation rate’.

Response 19: We have modified the sentence to say “The ReLC has a high phosphorylation rate, therefore it can only be determined by this method at 0°C (on ice) when the phosphorylation rate is reduced by 10-folds in comparison to that at 37°C.” (lines 315-318).

20)   Figure 6 is not cited in-text? And also is this showing the same sample as it progresses in the protocol?

Response 20: Thank you for pointing out this error. We have now referred to this figure in section 6 (line 545). Due to the order of appearance, this is now Figure 8. This is showing the same sample as it progresses. Lanes C-J show all purification steps after KES extraction. 

21)   Line 308: there is no lane P or O?

Response 21: This was a typo in the previous line 308. These two lanes should be lane I and lane J. We have now made the corrections in the legend (now Figure 6, lines 366-367, referring to lane I and lane J).

22)   Line 309: what’s the significance of phasic and tonic as they appeared suddenly?

Response 22: The significance of including phasic and tonic muscle samples is mainly to show that the method applies to different smooth muscle preparations. We have now modified the legend to clarify this point.

23)   Figure 8 also not cited in-text.

Response 23: Thank you for pointing out this error. Figure 8 is now Figure 7 (see Response 20 above), and is cited in section 3 (lines 323, 409, 410, and 460).

24)   Line 326-327: which method is preferred?

Response 24: It is not our intention to indicate that one is better than the other. It is up to the preference of the investigator.

25)   Line 516: compared ‘with’ not compared ‘to’.

Response 25: We corrected it as suggested (line 618).

26)   Line 541: the close bracket is missing.

Response 26: We have inserted the missing bracket behind “myosin” (line 654).

27)   Subheading for section 2: maybe can somehow indicate the evolvement towards the detection in ASM?

Response 27: We have added “(from gizzard to airway smooth muscle)” to the subheading for section 2.

28)   Section 5 and 6: not sure if I am missing the point/difference, but should section 6 come first (removal and purification) and then only section 5 (quantification)?

Response 28: We think section 5 should come before section 6 because the quantification of regulatory enzymes can only be done before the enzymes are isolated. During the isolation process, there is a loss of proteins that prevents an accurate estimate of quantity.

29)   Is it possible to have a figure for Section 6?

Response 29: We missed citing Figure 6 (which is now Figure 8, see Response 20 above) in section 6. The new Figure 8 is the figure for Section 6.

30)   Line 523: can you speculate or suggest what type of treatments?

Response 30: We have added a citation of our recent review to propose that inhibition of Rho-kinase coupled with therapeutic pressure oscillations to synergistically depolymerize myosin filaments can be a new treatment strategy for asthma (Wang et al. 2020b) (lines 627-629).

Reviewer 2 Report

This is a well-written review of a methodology; its optimization and potential applications is molecular understanding of AHR in asthma. The review presents a comprehensive, at times exhaustive, details on biochemical purification of myosin and associated regulatory proteins from various tissues. The objective appears to be down-scaling the technique to apply in ASM. The review contributes to the advancement of ASM biology by presenting an optimized methodology that could be used in studying biomechanical (i.e: evanescence) and molecular (in vitro phosphorylation/dephosphorylation) aspects of ASM shortening. I have two minor comments, addressing those I think will improve the quality of the article:

1.       The 1D PAGE images show named proteins. Probably it is best to label the molecular weight markers on those PAGE images. I understand these PAGE images may have been generated for other studies and used in this review. However, were any of those single bands ever been subject to a 2D gel to increase resolution of novel regulatory proteins?

2.       The current direction in ASM biology has moved into high resolution, high throughput methods such as MALDI-TOF and SILAC labeling for phosphoproteins. Do authors have any comment or insight into the downstream proteomic or Mass spec applications of these isolation methods?   

Author Response

Comments and Suggestions for Authors

This is a well-written review of a methodology; its optimization and potential applications is molecular understanding of AHR in asthma. The review presents a comprehensive, at times exhaustive, details on biochemical purification of myosin and associated regulatory proteins from various tissues. The objective appears to be down-scaling the technique to apply in ASM. The review contributes to the advancement of ASM biology by presenting an optimized methodology that could be used in studying biomechanical (i.e: evanescence) and molecular (in vitro phosphorylation/dephosphorylation) aspects of ASM shortening. I have two minor comments, addressing those I think will improve the quality of the article:

We would like to thank the reviewer for the comments and we have addressed both points. Please note the new line numbers refer to the line numbers when track changes are in place as required by the journal for revision submission.

  1. The 1D PAGE images show named proteins. Probably it is best to label the molecular weight markers on those PAGE images. I understand these PAGE images may have been generated for other studies and used in this review. However, were any of those single bands ever been subject to a 2D gel to increase resolution of novel regulatory proteins?

Response 1: Because of this and other reviewers’ comments we have decided to add to each figure a reference lane with protein bands labeled with protein names. We have chosen names instead of molecular weights because in many cases the position of the proteins is not strictly linear on the plots to the log of their MWs versus their relative migration positions. For example, caldesmon, which has a molecular weight of 135 kDa, migrates to 105 kDa level on SDS PAGE.  Another example is telokin, which has a molecular weight 18 kDa, migrates to 24 kDa.

This manuscript is a blending of a review and a method paper. We understand that this is appropriate for this special issue. Therefore, we prepared this method review in which we used previously unpublished figures to demonstrate the method in detail.  

The proteins were never subjected to 2D gel electrophoresis because their identification and characterization via purification are reliable using the method described in this manuscript. We agree with the reviewer that 2D gel may provide additional information and could be a powerful tool to be considered in the future.

  1. The current direction in ASM biology has moved into high resolution, high throughput methods such as MALDI-TOF and SILAC labeling for phosphoproteins. Do authors have any comment or insight into the downstream proteomic or Mass spec applications of these isolation methods?  

Response 2: The downstream proteomic or Mass spectrometry techniques can be certainly used to characterize the purified myosin and the isolated regulatory enzyme complex. In addition, these techniques have the potential to detect and discriminate structural and sequence alterations in the purified myosin and regulatory complex that are caused by diseases such as asthma. Although detailed discussions on these techniques are beyond the scope of this review, we have added some thoughts in section 7 “application in asthma research” (lines 631-637) to speculate on this point raised by the reviewer.  

Reviewer 3 Report

This manuscript provides interesting information regarding purification and characterization of myosin and associated proteins from bovine tracheal muscle.  While interesting, the review raises some questions:

1.                  The authors are reviewing different methods for purification of myosin, but the manuscript reads a bit like a review, but heavily details methods.  The authors even write into the abstract that it’s a purification protocol-based paper.  This manuscript may be out of scope for the journal in terms of the content of the paper.  It should either be a review or a methods paper, but looks like a blending of both. Also, it’s very biochemistry heavy, so maybe a biochemical methods journal would be best?

2.                  The bands referred to in the text of the manuscript and legends of the figures are not clear, as the blots are not labeled at all and specific bands are vaguely referred to.  The authors should go back and put actual labels on the blots so that the proteins that are being referred to are easily distinguished from the laddering effect seen on many of the gels.  Additionally, the blot in Figure 7 makes interpretation of the data difficult to make.

3.                  The authors state that Rho kinase protein expression is increased in ASM from asthmatics, but fail to also cite Koziol-White et al (2016) BJP where that group also noted increased Rho kinase activity in asthma-derived airway smooth muscle.  The contractility of the muscle from asthmatics augmented both from increased expression but also increased activity of Rho kinase.

Although the subject of the manuscript is potentially interesting within the field, the scope of the article make it much better suited for a biochemical methods paper rather than as a review in this journal.

Author Response

Comments and Suggestions for Authors

This manuscript provides interesting information regarding the purification and characterization of myosin and associated proteins from bovine tracheal muscle.  While interesting, the review raises some questions:

We would like to thank the reviewer for the comments and suggestions that have helped us improve our manuscript. Following the reviewer’s suggestions, we have made modifications to the manuscript and figures. Please note the new line numbers refer to the line numbers when track changes are in place as required by the journal for revision submission.

  1. The authors are reviewing different methods for the purification of myosin, but the manuscript reads a bit like a review, but heavily details methods.  The authors even write in the abstract that it’s a purification protocol-based paper.  This manuscript may be out of the scope of the journal in terms of the content of the paper.  It should either be a review or a methods paper, but looks like a blending of both. Also, it’s very biochemistry heavy, so maybe a biochemical methods journal would be best?

Response 1: As the reviewer pointed out, our manuscript is indeed a blending of a review and a method paper. We understand that this is appropriate for this special issue. As remarked by the academic editor, both review and method are welcome. We see that the reviewer also agrees that this manuscript can provide a significant contribution to the field.  We hope that this will give an important contribution to the special issue of Cells on Airway Smooth Muscle and Asthma.

  1. The bands referred to in the text of the manuscript and legends of the figures are not clear, as the blots are not labeled at all and specific bands are vaguely referred to.  The authors should go back and put actual labels on the blots so that the proteins that are being referred to are easily distinguished from the laddering effect seen on many of the gels.  Additionally, the blot in Figure 7 makes the interpretation of the data difficult to make.

Response 2: We have followed the reviewer’s suggestion and labeled all the gel images with arrows and the protein names of all the relevant bands. Figures 7 and 8 (now Figures 6 and 7) show a separation of the low molecular weight acidic proteins according to their charges on Urea Glycerol PAGE and are stained with Coomassie blue. We have now modified the legends hoping to make the interpretation more straightforward.

  1. The authors state that Rho kinase protein expression is increased in ASM from asthmatics, but fail to also cite Koziol-White et al (2016) BJP where that group also noted increased Rho kinase activity in asthma-derived airway smooth muscle.  The contractility of the muscle from asthmatics augmented both from increased expression but also increased activity of Rho kinase.

Response 3: We apologize for missing this important reference. We have added it to the text in section 7 (lines 618-620) and the reference list (#14).

Reviewer 4 Report

This manuscript by Lu Wang et al., presents a Purification of myosin from bovine tracheal smooth muscle and regulatory complexes such as MLCK and MLCP with a methodological review.  Personally, I do like the discussion more of PMLC and PMLK activation and regulation of myosin and actins with the calcium pathway.

Here is I have minor concerns.

Line 15: remove “_ “ after “:”

Figure 1: it is maybe better to change to format of each unit display such as MgATP:7.5 mM, KCI: 90 mM ……. Or 7.5 mM MgATP, 90 mM KCl, it is the same format as a second part 60 mM MgCl2.

Figure 4: Like figure 3 better to provide one line of figure 2 or like in figure 5 A-line whole bovine tracheal smooth muscle sample before the first washing sample.

Line 217: liquid N2 (LN2) changes to liquid Nitrogen (LN2).

Line 246: for better following the story Figs 4 and 5 change to Figs 4 line-L and 5 line-H.  

Figure 6: Authors provide whole tissues sample lines that will be better to follow reading to data.

Line 308: In figure 7, Hard to follow P and O.

Figure 8: It will be better if authors provide indicators such as an arrow or anything for phosphorylated protein, which gives readers a better understanding.

Line 372: within 10-15 sec to within 10-15 seconds.

In Figures 2 to 8:  Gel data showed estimated size (M.W) will be good for readers.

Table 1: change to 40,000 g x 16 hr; 10,5000 g x 2.5 hr; 12,000g x 45 min; Centrifuge 39,100 g x 30 min;  (2-20 mM) et al.

In references, authors need to make the same font for all reference formats.

Author Response

Comments and Suggestions for Authors

This manuscript by Lu Wang et al., presents a Purification of myosin from bovine tracheal smooth muscle and regulatory complexes such as MLCK and MLCP with a methodological review.  Personally, I do like the discussion more of PMLC and PMLK activation and regulation of myosin and actins with the calcium pathway.

We would like to thank the reviewer for the helpful comments. We have made all the corrections as suggested. Please note the new line numbers refer to the line numbers when track changes are in place as required by the journal for revision submission.  

Response 1: We understand that the reviewer would like to see more of the discussion on calcium and smooth muscle activation. We have added some details on the binding of calcium to troponin and calmodulin complex in the introduction (lines 67-72). However, we believe that an extensive discussion on calcium regulation of myosin and actin goes beyond the scope of this manuscript.

Here is I have minor concerns.

Line 15: remove “_ “ after “:”

Response 2: We have removed the underline (line 15).

Figure 1: it is maybe better to change to format of each unit display such as MgATP:7.5 mM, KCI: 90 mM ……. Or 7.5 mM MgATP, 90 mM KCl, it is the same format as a second part 60 mM MgCl2.

Response 3: We have made the suggested changes to Figure 1.

Figure 4: Like figure 3 better to provide one line of figure 2 or like in figure 5 A-line whole bovine tracheal smooth muscle sample before the first washing sample.

Response 4: Because of this comment and comments from other reviewers, we have modified Figure 2 which shows one lane of turkey gizzard myofibrils and the other lane showing bovine tracheal smooth muscle myofibrils. We have now added to each SDS PAGE gel figure, as a reference lane, the bovine tracheal smooth muscle myofibrils of the new Figure 2. All relevant protein bands are labeled with arrows and protein names.

Line 217: liquid N2 (LN2) changes to liquid Nitrogen (LN2).

Response 5: We have made the change (line 257).

Line 246: for better following the story Figs 4 and 5 change to Figs 4 line-L and 5 line-H.  

Response 6: We have made the change (line 290).

Figure 6: Authors provide whole tissues sample lines that will be better to follow reading to data.

Response 7: Because of a comment from Reviewer 1, this figure is now Figure 8. We have added a reference lane (see response 4 above).

Line 308: In figure 7, Hard to follow P and O.

Response 8: Reviewer 1 also made the same comment. This was a typo in the previous line 308. These two lanes should be lane I and lane J. We have now made the corrections in the legend (now Figure 6, lines 366-367, referring to lane I and lane J).

Figure 8: It will be better if authors provide indicators such as an arrow or anything for phosphorylated protein, which gives readers a better understanding.

Response 9: Figure 8 is now Figure 7. We have added arrows and labels for the phosphorylation state of each band.

Line 372: within 10-15 sec to within 10-15 seconds.

Response 10: We have made the change here (line 455) and throughout the manuscript by spelling out the words seconds, minutes, and hours.

In Figures 2 to 8:  Gel data showed estimated size (M.W) will be good for readers.

Response 11: This point was also raised by Reviewer 2. We have decided to add to each figure a reference lane with protein bands labeled with protein names. We have chosen names instead of molecular weights because in many cases the position of the proteins is not strictly linear on the plots to the log of their MWs versus their relative migration positions. For example, caldesmon, which has a molecular weight of 135 kDa, migrates to 105 kDa level on SDS PAGE.  Another example is telokin, which has a molecular weight 18 kDa, migrates to 24 kDa.

Table 1: change to 40,000 g x 16 hr; 10,5000 g x 2.5 hr; 12,000g x 45 min; Centrifuge 39,100 g x 30 min;  (2-20 mM) et al.

Response 12: We have made the changes everywhere applicable in Table 1. 

In references, authors need to make the same font for all reference formats.

Response 13: We have corrected this mistake and made all the references in the same font.

Round 2

Reviewer 3 Report

I feel that the authors addressed the concerns I had.